# [RE] GNNBoundary: Towards Explaining Graph Neural Networks through the Lens of Decision Boundaries

**Matei Nastase** *matei.nastase@student.uva.nl*

**Tyme Chatupanyachotikul** *tyme.chatupanyachotikul@student.uva.nl*

**Leonard Horns** *leonard.horns@student.uva.nl*

**Reviewed on OpenReview:** *https://openreview.net/forum?id=zLfLTHOdZW*

## Abstract

Graph Neural Networks (GNNs) can model complex relationships while posing significant interpretability challenges due to the unique and varying properties of graph structures, which hinder the adaptation of existing methods from other domains. To address interpretability challenges in GNNs, GNNBoundary was designed as a model-level explainability tool to provide insights into their overall behavior. This paper aims to thoroughly evaluate the reproducibility, robustness, and practical applicability of the findings presented in the original work by replicating and extending their experiments, highlighting both strengths and limitations while considering potential future improvements. Our results show that while the algorithm can reliably generate near-boundary graphs in certain settings, its performance is highly sensitive to hyperparameter choices and suffers from convergence issues. Furthermore, we find that the generated solutions lack diversity, often representing only a single region on the decision boundary, which limits their effectiveness in broader decision boundary analysis. All the code used throughout the research is publicly available on GitHub.

## 1 Introduction

Rapid advancements in Artificial Intelligence (AI) have led to its unprecedented integration across diverse domains, profoundly impacting companies, organizations, governments, and individuals (Rashid and Kausik, 2024). However, the widespread adoption of AI-driven algorithms raises pressing concerns regarding their reliability, accountability, and ethical implications. Consequently, research efforts have shifted focus from merely improving performance to developing tools that ensure AI systems are transparent, robust, and aligned with ethical standards (Bibi, 2024).

Graph Neural Networks (GNNs) exemplify this challenge, offering an unparalleled ability to model complex relationships in graph data, but posing significant interpretability hurdles due to the unique and varying properties of graph data (Wang and Shen, 2023), which prevent the adaption of most existing methods from other domains (Yuan et al., 2022). These challenges are particularly concerning given GNNs' deployment in applications where trustworthiness is critical (Wu et al., 2022a), such as molecular biology (Zhang et al., 2021), fraud detection (Cheng et al., 2022), and recommendation systems (Wu et al., 2022b). Despite their growing prominence, research on GNN explainability remains limited and mainly covers instance-level methods (Wang and Shen, 2023). Building on their prior work: GNNInterpreter (Wang and Shen, 2023), *GNNBoundary: Towards Explaining Graph Neural Networks Through the Lens of Decision Boundaries* (Wang and Shen, 2024) further addresses this gap by proposing a model-level explainability method aimed at demystifying GNN decision-making through the generation and analysis of graphs near the decision boundaries of a trained model.

## 2  Scope of reproducibility

This study aims to thoroughly evaluate the reproducibility, robustness, and practical applicability of the findings presented in the original paper (Wang and Shen, 2024), by replicating and extending their experiments. The main ideas of their work are summarised in the following claims:

1. GNNBoundary can reliably identify pairs of adjacent classes within the embedding space generated by a GNN.

2. The GNNBoundary algorithm generates faithful near-boundary graphs and is applicable to any GNN using the message-passing framework, regardless of the scale of the dataset.

3. The generated near-boundary graphs are essential for analysing the model's decision-making process. Their analysis involves three key metrics:
    (a) Explaining the model's susceptibility to specific misclassifications through the *boundary margin*.
    (b) Calculating the *boundary thickness* to evaluate the model's robustness to perturbations, such as adversarial attacks.
    (c) Using the *boundary complexity* to assess how well the model represents the data.

4. The adaptive loss function proposed by the authors enables faster convergence of the GNNBoundary algorithm and lowers the risk of local minima compared to the standard cross-entropy loss.

We can verify Claim 2, but Claims 1, and 4 are only partially verifiable. Claim 3 is unverifiable for us, and we find evidence that contradicts it. We perform additional experiments on the authors' work to highlight the method's strengths and weaknesses, and to reflect on potential improvements. Our contributions are as follows:

- We show that GNNBoundary can consistently generate near-boundary graphs in certain settings but exhibits convergence issues and is highly dependent on the choice of hyperparameters.

- We investigate the limitations of GNNBoundary and demonstrate that the solutions found by the algorithm lack diversity and only represent a small region of the decision boundary.

- We apply GNNBoundary to a new architecture, showing that the method struggles with more complex models, but can still produce useful results.

## 3  Prerequisites

GNNBoundary (Wang and Shen, 2024) is a novel model-level explainability tool that provides insights into the overall behavior of GNNs. Unlike prior instance-level explainability methods (Luo et al., 2020; Ying et al., 2019; Vu and Thai, 2020), which focus on individual predictions, this tool analyses broader decision-making patterns across an entire dataset. To develop the necessary background knowledge, we first review some key concepts in graph theory, graph neural networks, and the closely related GNNInterpreter algorithm (Wang and Shen, 2023).

### 3.1  General theory and notations

For consistency, we use the same mathematical notations as in the original papers. We represent a graph $G = (\mathcal{V}, \mathcal{E})$ as a set of $N$ nodes, $\mathcal{V}$, and a set of $M$ edges, $\mathcal{E} \subseteq \mathcal{V} \times \mathcal{V}$. The node pairs in $\mathcal{E}$ may be ordered or unordered, distinguishing directed and undirected graphs. In memory, the graph is usually represented by a binary adjacency matrix $\mathbf{A} \in \{0,1\}^{N \times N}$ where $a_{i,j} = 1$ only if there is an edge from node $v_i$ to $v_j$ and 0 otherwise.

GNNs (Scarselli et al., 2009) leverage message-passing mechanisms for learning node representations. A GNN with $L$ layers consists of an embedding function $\phi$ (first $K$ layers) and a scoring function $\eta$ (last $L - K$

layers before softmax). We denote their sub-functions up to a layer $l$ as $\phi_l$ and $\eta_l$, with the hidden node representations given by $\mathbf{H}^{(l)} = \phi_l(G) \in \mathbb{R}^{N \times D^{(l)}}$, where $D(l)$ is the number of dimensions of the embedding.

At each layer, the network partitions the embedding space into $C$ decision regions $\{\mathcal{R}_c^{(l)} \mid c \in [1, C]\}$, where $\mathcal{R}_c = \mathcal{R}_c^{(0)}$ represents the decision region for class $c$ in the input space. The classifier assigns class $\hat{c}$ to the graph if $\hat{c} = \arg\max_c f_c(G)$, or in other words, $G \in \mathcal{R}_{\hat{c}}^{(L)}$. The decision boundary between two classes $\mathcal{B}_{c_1 \| c_2}$ is defined as the region in the embedding space where the probability of a graph being assigned to either class $c_1$ or class $c_2$ is exactly 0.5.

## 3.2 GNNInterpreter

A previous approach for model-level explainability in GNNs is the GNNInterpreter (Wang and Shen, 2023). Similar to the GNNBoundary algorithm, GNNInterpreter employs a Monte Carlo strategy to learn the parameters of a distribution over the input space, enabling the sampling of graphs that maximise the model's confidence in its prediction for a given class $c$. While the underlying methodology is the same for both algorithms, the learning objective is very different, requiring different criteria to facilitate convergence. GNNInterpreter utilises an additional objective to ensure that the generated data belongs to the true data distribution. This cannot be applied to GNNBoundary as datapoints belonging to two classes at once should not be part of the data distribution in typical classification problems.

# 4 Methodology

To reproduce the experiments, we make use of the official implementation published by the authors. Their repository provides access to an implementation of GNNBoundary, datasets, environment details, and model checkpoints. We identify and resolve some discrepancies between the paper and the implementation, along with minor bug fixes, and expand the codebase by implementing the missing margin, thickness, and complexity metrics, as well as our additional experiments. Our updated version is available in this repository[1].

## 4.1 GNNBoundary Algorithm

**Identifying Adjacent Classes**  We identify adjacent classes (sharing a decision boundary) using the adjacency score proposed by Wang and Shen (2024), which measures the probability that graphs sampled from the respective classes form an adjacent pair. Two classes are considered adjacent if their score exceeds a set threshold, indicating that most of the straight paths between the decision regions of the two classes do not cross any other class in the embedding space.

**Generating boundary graphs**  To generate graphs near a decision boundary, we learn a distribution over which graphs best represent this boundary, assuming independence between both the edges (Gilbert random graph, Gilbert (1959)) and node features (Wang and Shen, 2024). Following their method, we optimise this distribution by using Stochastic Gradient Descent (SGD) to minimise their novel objective function and applying a continuous relaxation (Maddison et al., 2017) to the edges and node features to enable gradient flow. This objective function is a modified version of the cross-entropy loss with target probabilities of 0.5 for both boundary classes, where the logits of the boundary classes are never minimised during training. We approximate the integral over our distribution through Monte Carlo integration. Additionally, we add both L1 and L2 regularisation on the parameters of the edge distribution, as well as a budget penalty on the number of edges following Wang and Shen (2023). Their method dynamically increases the weight for this budget penalty when the main convergence criterion is met, but the expected number of edges in the learnt distribution exceeds a predefined threshold, preventing convergence. Conversely, it iteratively reduces the weight if the criterion is no longer satisfied. We consider the algorithm converged, if the expected class probabilities for both boundary classes are within the range $[0.45, 0.55]$.

---

[1] https://github.com/leonardhorns/GNNBoundary

### 4.1.1 Boundary Analysis

Wang and Shen (2024) propose that a set of sampled boundary graphs for a class pair $(c_1, c_2)$ can be used to analyse the properties of the decision boundary by approximating $\mathcal{B}_{c_1||c_2}$. Following Wang and Shen (2024), we implement three quantitative measures for this analysis.

**Boundary margin**   The boundary margin measures the minimum distance between a classifier's intermediate representation and the decision boundary, providing insight into the model's generalisation and stability to input perturbations (Elsayed et al., 2018). The asymmetric margin boundary equation formulated by Yang et al. (2020) is:

$$\Phi\left(f, c_1, c_2\right) = \min_{\left(G_{c_1}, G_{c_1||c_2}\right)} \left\| \phi\left(G_{c_1}\right) - \phi\left(G_{c_1||c_2}\right) \right\| \tag{1}$$

As done by Wang and Shen (2024), we obtain the boundary margin by selecting $\phi_l(G)$ to be the output embedding produced by the graph pooling layer, which aligns with standard practices for interpreting deep neural networks (Bajaj et al., 2021). To obtain the boundary margin, we calculate the minimum distance over all pairings of dataset graphs belonging to class $c_k$ and a set of sampled boundary graphs.

**Boundary thickness**   Boundary thickness measures the expected distance between a level of predicted class probability in a class' decision region and a corresponding boundary. Yang et al. (2020) show that thicker boundaries enhance robustness, while thinner ones lead to overfitting. The equation for asymmetric boundary thickness is:

$$\Theta\left(f, \gamma, c_1, c_2\right) = \mathbb{E}_{\left(G_{c_1}, G_{c_1||c_2}\right) \sim P} \left[ \left\| \phi\left(G_{c_1}\right) - \phi\left(G_{c_1||c_2}\right) \right\| \int_0^1 \mathbb{1}_{\gamma > \sigma(\eta(h(t)))_{c_1} - \sigma(\eta(h(t)))_{c_2}} \, dt \right] \tag{2}$$

where P is a distribution over pairs of points $\left(G_{c_1}, G_{c_1||c_2}\right) \sim P$. We use $h(t) = (1-t) \cdot \phi\left(G_{c_1}\right) + t \cdot \phi\left(G_{c_1||c_2}\right)$ for $t \in [0,1]$, and $\gamma = 0.75$ as done by Wang and Shen (2024). We approximate the integral term in the equation through discretisation: $\frac{1}{\kappa} \sum_{t=0}^1 \mathbb{1}_{\gamma > \sigma(\eta(h(t)))_{c_1} - \sigma(\eta(h(t)))_{c_2}}$ where $\Delta t$ is 0.02, and $\kappa = 50$.

**Boundary complexity**   Complexity is a measure of the model's generalisability, where a high decision boundary complexity might indicate overfitting (Guan and Loew, 2020). The generated boundary graphs approximate the decision boundary and can be used to measure boundary complexity by:

$$\Gamma\left(f, c_1, c_2\right) = H\left(\boldsymbol{\lambda}/\|\boldsymbol{\lambda}\|_1\right) / \log D = \left( -\sum_i \left(\lambda_i/\|\boldsymbol{\lambda}\|_1\right) \log\left(\lambda_i/\|\boldsymbol{\lambda}\|_1\right) \right) / \log D \tag{3}$$

where $\boldsymbol{\lambda}$ denotes the eigenvalues of the covariance matrix of boundary graphs $G_{c_1||c_2} \in \mathcal{B}_{c_1||c_2}$. Following Wang and Shen (2024), we work with the sample covariance of the last hidden layer representations, $\eta_{L-1}(G_{c1||c2})$, as this complexity measure applies only to linearly separable boundaries.

## 4.2 Datasets

We evaluate the GNNBoundary algorithm on all three datasets used in the original work: Motif, Collab, and Enzymes, as well as one additional dataset. For each dataset, we reserve 10% of the samples for model evaluation. It is worth mentioning that dataset sizes reported here (Table 1) are slightly higher than those in Wang and Shen (2024), likely due to updates since the original study.

The *Motif* dataset[2] is a synthetic dataset with classes: House, House-X, Complete-4, and Complete-5, which indicate a specific pattern that appears in each graph within this class. The *Collab* dataset[3] is a real-world dataset containing graphs representing collaboration networks of researchers from three related physics fields: High Energy, Condensed Matter, and Astro. Notably, more than half of the samples belong to the High

---

[2]https://networkx.org/documentation/stable/reference/generators.html

| Dataset | Nodes | Edges | Classes | Examples | Node feature |
|---------|-------|-------|---------|----------|--------------|
| Motif | 57.07 | 77.08 | 4 | 11,531 | Categorical |
| Collab | 74.49 | 2457.78 | 3 | 5,000 | None |
| Enzymes | 32.63 | 62.14 | 6 | 600 | Categorical |
| Reddit | 429.63 | 497.75 | 2 | 2,000 | None |

Table 1: Dataset statistics. For the number of nodes and edges, we report the mean across all examples.

Energy class. Lastly, the *Enzymes* dataset[3] is another real-world dataset consisting of protein structures that can be divided into different classes.

To expand our investigation for Claim 2, we also apply GNNBoundary to the *Reddit-Binary* dataset[3] (Reddit). It consists of graphs representing the user interactions in Question-Answer or Discussion threads. Despite having only two classes, its significantly larger graph size (see Table 1) presents a unique challenge, allowing us to assess GNNBoundary's effectiveness on larger graphs without the added complexity of multiple decision regions.

### 4.3 GNN Architectures

All original experiments follow the same architecture, consisting of initial graph convolutional layers (Kipf and Welling, 2017), a pooling layer that applies sum and mean-pooling, a fully connected layer with a ReLU activation, and a final linear classifier. The model sizes are chosen to optimise performance, with specific configurations for each dataset: 3 graph-convolutional layers (embedding size of 6) for the Motif dataset, 5 layers (embedding size of 64) for Collab, and 3 layers (embedding size of 32) for Enzymes. For the additional Reddit-Binary dataset, we also use 5 graph convolutional layers with an embedding size of 64. In our evaluations, we analyse results from both the provided checkpoints and models we train ourselves (see Appendix B for training details). Finally, we test GNNBoundary on a Graph Attention Network (GAT) (Veličković et al., 2018) architecture by replacing the graph convolutional layers with graph attention layers, while keeping the number of layers and embedding size the same.

### 4.4 Hyperparameters

We base our hyperparameters on those reported in the original paper. However, as not all were documented, we also refer to their published demo notebooks. Notably, some hyperparameters differ between the paper and the demo, suggesting potential variations in the original experiments. For our additional investigations, we manually tune GNNBoundary's hyperparameters to ensure convergence, such as accommodating for the larger graph size in the Reddit dataset. However, we refrain from doing a full hyperparameter search because, between convergence rate and resulting class probabilities, the concrete objective for the search is unclear. Additionally, robustly approximating their performance would require a large number of runs per parameter combination, which is infeasible with limited resources. A full list of hyperparameters for both the original and additional experiments is provided in Appendix B.

### 4.5 Experimental setup and code

A key difference between our approach and the original method is that we sample directly from the dataset when needed, whereas the authors frequently made use of GNNInterpreter to produce graph samples. We choose this approach to ensure the validity of the claims is independent of the sampling source (Wang and Shen, 2024) and to evaluate GNNBoundary as a standalone method.

**Adjacency between classes** For each model, we establish the pairs of adjacent classes using the previously mentioned method from Section 4.1. We set the same threshold of 0.8 adjacency for every experiment. To

---

[3]https://chrsmrrs.github.io/datasets/

mitigate stochasticity and fluctuations of borderline scores inherent to the method, we repeat the experiment 10 times and report mean adjacency scores for robustness.

**Boundary graphs**  To verify the capability of successfully learning a boundary graph distribution, we test the algorithm on all adjacent class pairs of each respective dataset. For each pair, we run the algorithm 1,000 times, for a maximum of 500 iterations each. For every successful run (see Section 4.1), we sample 500 graphs from the learnt distribution and compute the mean and standard deviation over their predicted class probabilities. We report the convergence rate, as well as both the statistics for the best run, and their average across runs. For every pair of classes $(c_1, c_2)$, we select as the best run, the sampler that minimizes:

$$\sum_{c \in \{c_1, c_2\}} |\text{mean}(p(c)) - 0.5| + \text{std}(p(c)) \tag{4}$$

**Baseline**  The methodology for generating baseline boundary graphs was obtained from Wang and Shen (2024), which involves connecting a random edge between a randomly sampled pair of graphs: $G_1 \in \mathcal{R}_{c_1}$, $G_2 \in \mathcal{R}_{c_2}$. The rationale behind this idea is that the boundary graph should contain discriminative features for both of the adjacent classes. We sample 500 graphs per class pair for the evaluation of GNNBoundary.

**Boundary analysis**  To generate the necessary boundary graphs, we sample from the graph distribution of the best converged GNNBoundary run. Since these evaluation metrics are influenced by the quality of the generated graphs, we only use graphs for which the predicted probabilities of the adjacent class pair lie within the range $[0.45, 0.55]$. We use the same 500 sampled boundary graphs to calculate each metric.

### 4.6  Computational requirements

Our experiments have modest hardware requirements, as both the models and datasets are of moderate size. All experiments can be reproduced using an Intel Core i7-10870H processor (8 cores, 16 threads, 2.2 GHz base, 5.0 GHz boost). However, to accelerate computationally intensive tasks (such as training 1,000 samplers per class pair) we utilized an NVIDIA A100 GPU. The total computation time for our experiments was approximately 30 hours. During this period, we estimate an energy consumption of 4.22 kWh, resulting in 1.87 kg of $CO_2$ emissions, roughly equivalent to driving a car for 8 km.

## 5  Results

### 5.1  Results reproducing original paper

| Dataset | Original Paper | Provided Checkpoints | Retrained |
|---------|----------------|----------------------|-----------|
| Motif   | 0.99           | 0.95                 | 1.00      |
| Collab  | 0.74           | 0.85                 | 0.78      |
| Enzymes | 0.52           | 0.73                 | 0.53      |

Table 2: Model test set accuracies on each dataset. Columns show accuracies reported in the original paper, the accuracies of the official models' checkpoints, and our own retrained checkpoints.

**Model training**  Analysing Table 2, we observe significant discrepancies between the reported model accuracies and the provided checkpoints, making the exact reproduction of the authors' results challenging due to the nature of the GNNBoundary algorithm. These differences in accuracy may stem from unreported factors such as initialisation methods or random seed choices that lead to more favorable test splits, particularly for the small-sized Enzymes dataset. Notably, training the models from scratch with the given hyperparameters yields accuracies more aligned with the original results. However, since our primary objective is to replicate the authors' analysis, we mainly focus on the results obtained using the provided checkpoints in the following sections, while additional findings from our trained models are detailed in Appendix C.

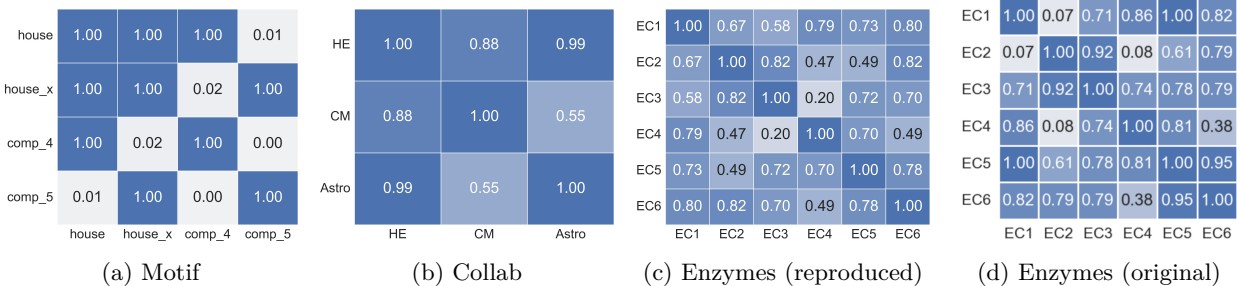

(a) Motif      (b) Collab      (c) Enzymes (reproduced)      (d) Enzymes (original)

Figure 1: Adjacency scores for each model trained on the respective datasets.

Following the authors' approach, we first compute the adjacency scores for all classes across datasets and consider adjacent only the pairs that surpass the threshold of 0.8 as presented in Figure 1. We successfully reproduce the same pairs of adjacent classes for Motif and Collab, but not for Enzymes. Although adjacent classes should generally share some features in the input space, their representation in the embedding space may vary significantly depending on the initialisation of the model and the local minimum to which it converges. For this reason, obtaining different pairs does not invalidate the correctness of the algorithm. On the other hand, we could not find any correlation between the adjacency scores and the performance of the GNNBoundary algorithm. Moreover, the presence of two outlier pairs: EC2 & EC3 and Condensed Matter & Astro (see Table 3), makes Claim 1 only partially verifiable.

| Dataset | Class Pair | | Complexity | | GNNBoundary | Baseline |
|---|---|---|---|---|---|---|
| | $c_1$ | $c_2$ | Orig. | Repr. | $p(c_1)$ / $p(c_2)$ | $p(c_1)$ / $p(c_2)$ |
| Motif | House | HouseX | 6.55e-8 | 0.200 | 0.489 (0.039) / 0.511 (0.039) | 0.054 (0.926) / 0.166 (0.182) |
| | House | Comp4 | 6.55e-8 | 0.122 | 0.504 (0.002) / 0.496 (0.002) | 0.291 (0.699) / 0.295 (0.306) |
| | HouseX | Comp5 | 2.57e-7 | 0.163 | 0.490 (0.046) / 0.510 (0.046) | 0.745 (0.018) / 0.339 (0.096) |
| Collab | HE | CM | 0.0463 | 0.276 | 0.485 (0.015) / 0.464 (0.025) | 0.847 (0.131) / 0.310 (0.295) |
| | HE | Astro | 0.0246 | 0.174 | 0.495 (0.017) / 0.478 (0.018) | 0.363 (0.629) / 0.460 (0.466) |
| | CM | Astro | – | 0.031 | 0.492 (0.047) / 0.503 (0.048) | 0.057 (0.281) / 0.208 (0.436) |
| Enzymes | EC1 | EC6 | 0.1492 | 0.218 | 0.457 (0.080) / 0.465 (0.074) | 0.135 (0.274) / 0.135 (0.274) |
| | EC2 | EC3 | 0.2325 | – | – / – | 0.118 (0.201) / 0.230 (0.317) |
| | EC2 | EC6 | – | 0.012 | 0.480 (0.040) / 0.481 (0.040) | 0.176 (0.354) / 0.298 (0.384) |
| Reddit | Q&A | Discuss | – | 0.182 | 0.462 (0.076) / 0.538 (0.076) | 0.807 (0.193) / 0.237 (0.237) |

Table 3: GNNBoundary results per dataset and class pair. For complexity, we report the original and our reproduced values. We report mean and standard deviation over sampled graphs for the class probabilities and use samples from the best converging run for GNNBoundary. All shown class pairs are adjacent, aside from (CM, Astro).

**Generating Boundary Graphs** For the datasets Motif and Collab, we are mostly able to produce the results of Wang and Shen (2024). Except for the class pair House-X & Comp-5 the convergence rates of GNNBoundary are very high (see Table 4) and the best runs for each respective class pair consistently generate graphs with boundary class probabilities close to 0.5 (see Table 3). The generated graphs also consistently outperform our baseline, which performs similarly to the results reported by Wang and Shen (2024). These results directly support Claim 2.

For the Enzymes dataset, the convergence rates of GNNBoundary are drastically lower, and we are not able to achieve convergence at all for the class pair EC2, EC3 (see Table 4). Furthermore, the generated boundary graphs for the best runs seem to resemble the boundary less accurately (see Table 3), although they still outperform the baseline. In contrast, Wang and Shen (2024) were able to achieve high convergence rates for all 6 adjacent class pairs that they identified for their model. We likely observe different results because we have not been able to find good GNNBoundary hyperparameters for this dataset. Nonetheless, our results

| Dataset | Class Pair | | Convergence | | Run Means | Run Std. Devs. |
|---------|------------|--------|-------------|-------|-----------|----------------|
| | $c_1$ | $c_2$ | Orig. | Repr. | $p(c_1)$ / $p(c_2)$ | $p(c_1)$ / $p(c_2)$ |
| Motif | House | HouseX | 0.86 | 0.940 | 0.512 (0.042) / 0.487 (0.042) | 0.156 (0.062) / 0.158 (0.063) |
| | House | Comp4 | 1.00 | 0.833 | 0.516 (0.031) / 0.483 (0.031) | 0.051 (0.024) / 0.051 (0.024) |
| | HouseX | Comp5 | 0.76 | 0.096 | 0.503 (0.041) / 0.497 (0.041) | 0.110 (0.046) / 0.110 (0.046) |
| Collab | HE | CM | 1.00 | 0.999 | 0.486 (0.027) / 0.475 (0.028) | 0.102 (0.036) / 0.105 (0.036) |
| | HE | Astro | 1.00 | 0.988 | 0.481 (0.027) / 0.492 (0.029) | 0.072 (0.024) / 0.085 (0.028) |
| | CM | Astro | – | 0.982 | 0.499 (0.040) / 0.493 (0.039) | 0.152 (0.051) / 0.150 (0.051) |
| Enzymes | EC1 | EC6 | 0.75 | 0.002 | 0.460 (0.005) / 0.464 (0.001) | 0.090 (0.014) / 0.077 (0.006) |
| | EC2 | EC3 | 0.74 | 0.000 | – / – | – / – |
| | EC2 | EC6 | – | 0.062 | 0.470 (0.028) / 0.485 (0.030) | 0.094 (0.060) / 0.084 (0.057) |
| Reddit | Q&A | Discuss | – | 0.222 | 0.480 (0.041) / 0.520 (0.041) | 0.167 (0.031) / 0.167 (0.031) |

Table 4: GNNBoundary results per dataset and class pair. For convergence, we report the original and our reproduced values. In Table 3, we only report the mean and standard deviation (std) of sampled graphs for the best run. Here, we look at the behaviour of these statistics across all converging runs. The columns *Run Means* and *Run Std. Devs.* display the mean and standard deviation of graph samples per run in the format 'mean (std)'. All shown class pairs are adjacent, aside from (CM, Astro).

show that the algorithm can struggle with more complex datasets and is sensitive to the hyperparameter configuration.

Convergence is also highly dependent on both the dataset and the class pair. For example, the convergence rate of the class pair House-X & Comp-5 is drastically lower than for the other pairs in this dataset (see Table 4), although the adjacency scores are equally good (see Figure 1). Additionally, these average class probabilities are produced by the best converging runs for each class. We can see from Table 4 that the mean class probability for sampled boundary graphs varies significantly across multiple runs, indicating that for several runs the generated boundary graphs are biased towards one class. The average standard deviation (Table 4) is also a lot higher than in our best runs, producing more graphs that do not closely represent the boundary.

**Boundary Analysis**   In this section, Claim 3 is further investigated. The different boundary metrics are calculated and shown in Appendix D. The numbers vary from the original paper, however, this is expected as we suspect the provided model checkpoints to be different from the model checkpoints used originally by Wang and Shen (2024).

Furthermore, Wang and Shen (2024) claim that the boundary margin is a measurement of the robustness of boundaries to input perturbations, which is related to the risk of misclassification (Claim 3.a); and have shown that the relative size of the boundary margin of adjacent classes can be used to evaluate the relative likelihood of misclassification. However, we are unable to replicate this finding, as we cannot find any correlation between the boundary margin and the misclassification rate. Further analysis of this can be found in Appendix A.

For boundary complexity, Wang and Shen (2024) claim that this metric can be used to analyse the complexity of the decision making rule. As our datasets have very different characteristics, and we use varyingly powerful models, we would expect the learnt decision boundaries to be very different. The authors anticipated that the decision boundary complexity of the Enzyme's model should be the most complex, as the classification of real-world datasets with multiple classes should be more challenging than synthetic datasets with fewer classes, which is supported by their numbers. However, in our experiments the values for the boundary complexity lie within a similar range for all three datasets, and there is no visible correlation (see Table 3). Therefore, we are unable to accept Claim 3.c.

These results make us question the suitability of GNNBoundary samples for the computation of these metrics, which is why we refrain from further discussion of boundary thickness here. Instead, we investigate how well samples represent the decision boundary in Section 5.2.

## 5.2 Results beyond original paper

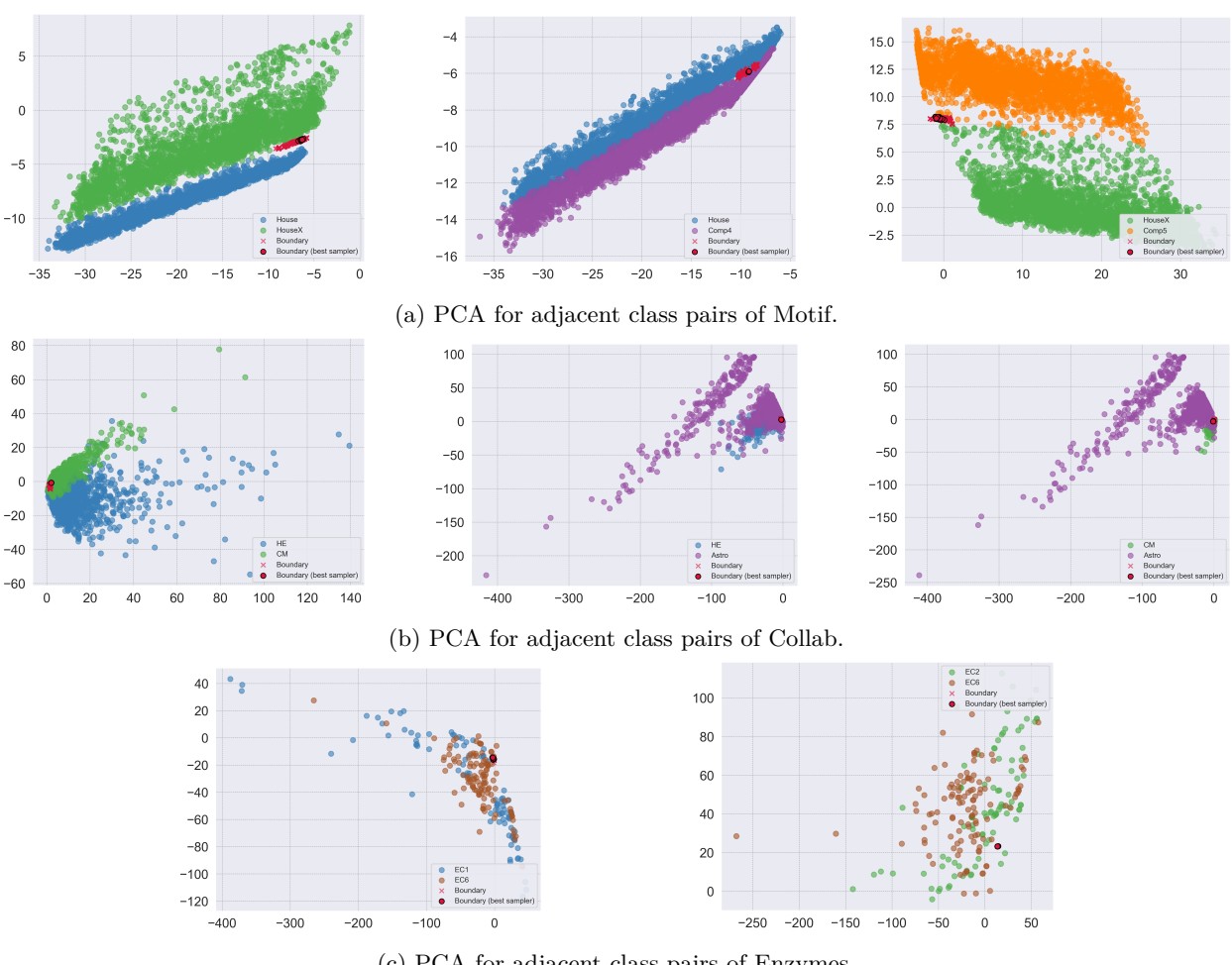

(a) PCA for adjacent class pairs of Motif.

(b) PCA for adjacent class pairs of Collab.

(c) PCA for adjacent class pairs of Enzymes.

Figure 2: Principle Component Analysis (PCA) of the Graph embedding space for the original datasets. We plot the Graph embeddings $\eta_{L-1}(G)$ of dataset graphs and generated boundary graphs $G$ projected onto the first two principal components for all adjacent class pairs. For the generated boundary graphs, we sample 50 graphs from 10 different converged runs and highlight the result from the best run.

**Qualitative analysis of the generated graphs** In this section, we conduct further investigations to understand why we were unable to reproduce Claim 3. A reason could be that the graph samples have a low variance, preventing them from adequately capturing the full landscape of the boundary between the two classes. To verify this, we project the graph embeddings of adjacent class pairs and their corresponding boundary graph embeddings onto the first two principal components using Singular Value Decomposition (SVD) to visually assess their diversity. For each adjacent class pair, we randomly select 10 different converging runs (including the best one), and sample 50 graphs from each of the samplers.

As shown in Figure 2, each boundary graph distribution from the best sampler converges to a single region, with graphs being generated only from its immediate vicinity. Although these graphs seem to lie at the boundary between the two classes, which supports Claim 2, they only represent a small region of it. This is particularly evident in Figure 2a, where the simplicity of the Motif dataset and the lower dimensionality of the model's embeddings show the separation between classes and the boundary. This introduces a significant bias in the computation of the suggested boundary metrics (margin, thickness, and complexity), and explains why the values we obtained were inaccurate. Furthermore, we have observed that the variation in generated

graphs are minimal, and all runs produce similar graphs. Utilizing all the graphs from the different samplers does still not yield an accurate representation of the boundary, potentially highlighting the limitation of this method.

To further support this point, we examine the distribution of the graph distribution parameters. The distribution has two types of parameters: $\theta_{ij}$ and $p_{ik}$. $\theta_{ij}$ represents the probability of sampling an edge between nodes $i$ and $j$, while $p_{ik}$ denotes the probability of sampling the discrete feature $k$ for a node $i$.

The parameter distributions of the learnt graph distributions are shown in Appendix F. We observe that the edge and node feature probabilities are highly skewed toward 0 or 1 for all class pairs across all datasets. This indicates that iteratively sampling from this distribution will consistently produce similar graphs, which can be verified by visualizing the samples (see Appendix G). The evidence presented demonstrates that conducting a comprehensive analysis of the entire boundary using a single GNNBoundary sampler is infeasible, as the method fails to capture the full diversity of the boundary. Therefore, we refute Claim 3.

| Dataset | Class Pair | | Convergence | GNNBoundary | Baseline |
|---|---|---|---|---|---|
| | $c_1$ | $c_2$ | | $p(c_1)$ / $p(c_2)$ | $p(c_1)$ / $p(c_2)$ |
| Motif | House | HouseX | 0.016 | 0.441 (0.127) / 0.556 (0.126) | 0.081 (0.200) / 0.852 (0.306) |
| | HouseX | Comp5 | 0.529 | 0.486 (0.045) / 0.496 (0.044) | 7.02e-5 (5.54e-4) / 1.000 (0.000) |
| | HouseX | Comp4 | – | – / – | 0.080 (0.242) / 0.920 (0.242) |
| | Comp4 | Comp5 | – | – / – | 0.026 (0.120) / 0.022 (0.139) |

Table 5: GNNBoundary results and convergence rates for the adjacent class pairs of the GAT model trained on Motif. For class probabilities, we report the mean and standard deviation over sampled graphs using samples from the best converging run for GNNBoundary. All values are rounded to three decimal places. Additionally, the development of the mean and standard deviation across runs can be found in Appendix E.

**Graph-Attention-Network**   Our previous experiments have demonstrated that the GNNBoundary algorithm can successfully generate graphs near the boundary between two classes, as Claim 2 asserts. However, we have relied solely on the GCN architecture so far, which raises concerns about the generalisability of the method. To address this, we apply the GNNBoundary to GAT, a model with comparable performance on benchmarks (Dwivedi et al., 2022), but a distinct message-passing mechanism.

Achieving convergence for GNNBoundary on the GAT network proves more challenging than on GCNs due to persistent exploding gradients, which underscore the increased complexity of a GAT's latent space. To mitigate this, we modify some of the hyperparameters (see Appendix B), stabilizing the learning curves across experiments. For the class pairs where GNNBoundary converges, we achieve samplers of comparable quality (see Table 5) to those trained on the GCN architecture, supporting the claim that the method can be extended to other graph networks.

However, despite these adjustments, the convergence rates of the algorithm remain low, and for half of the class pairs, GNNBoudnary fails to converge even after extensive hyperparameter tuning. This highlights the challenges that arise when applying GNNBoundary to more complex message-passing schemes and emphasizes the need for a deeper understanding of the method's optimisation process, which could be explored in future work.

Additionally, an interesting phenomenon emerged when tracking the class probability progression (see Figure 3) for the pairs House & House-X and House & Comp-5 that share a common class (House), but have very different convergence rates of 2.1% and 49.2% respectively. We attribute this to the latent space structure, where House, House-X, and Comp-5 might form a triplet structure rather than two distinct adjacent pairs. This leads to a conflict for our objective: increasing House-X's probability inadvertently raises Comp-5's, preventing convergence. Such scenarios will be even more prevalent in datasets with more classes, which raises concerns about the scalability of the method in this regard, and explains our difficulties in reproducing the results of Wang and Shen (2024) for the Enzymes dataset.

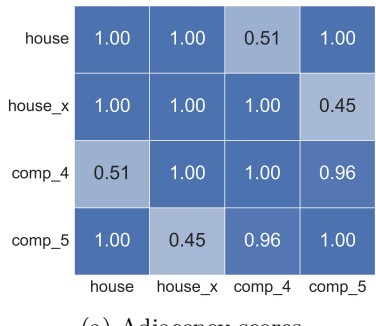

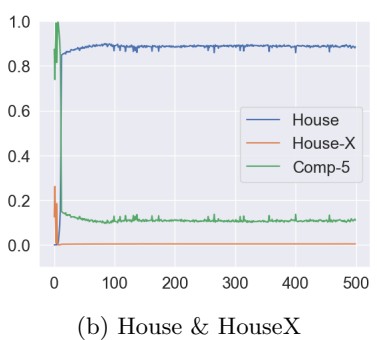

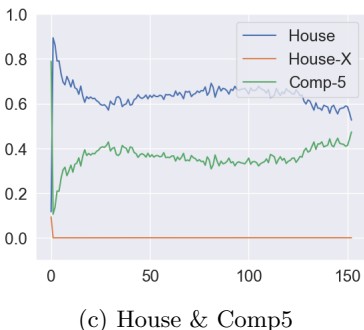

(a) Adjacency scores

(b) House & HouseX

(c) House & Comp5

Figure 3: Adjacency scores and development of class probabilities during training for the GAT on Motif. The class probabilities are shown for both adjacent pairs, namely House & HouseX, and House & Comp5.

**Reddit-Binary**  To assess the scalability of GNNBoundary in terms of the graph size, we introduce the Reddit-Binary dataset. Its samples contain significantly more nodes compared to the datasets used by Wang and Shen (2024) (see Table 1), which drastically increases the complexity of the learnt graph distribution. Moreover, we select a binary classification problem to evaluate the algorithm's performance on more complex data without encountering the complications discussed in the previous section.

In this new setting, we conduct additional hyperparameter tuning, which yield moderate results. While the convergence rate (see Table 4) is higher than that observed for the Enzymes dataset, it remains significantly lower than for the more successful datasets. The boundary complexity values (see Table 3) are consistent with those observed for other datasets. In contrast, the mean class probabilities per run are slightly worse, accompanied by a higher standard deviation. This outcome is anticipated, given the relatively lower number of convergent runs and the higher variance inherent to sampling from a larger graph distribution. These results support the idea that the increased scale of the dataset does not hinder the ability of the GNNBoundary to converge, which aligns with Claim 2. However, as shown in our experiments, scaling up the dataset does introduce several challenges, indicating that additional training strategies may still be needed to address these issues.

## 6  Discussion

Based on our results, we can only partially accept the claims made by the authors. Their GNNBoundary algorithm drastically outperforms a naive approach, such as the proposed baseline, and can consistently generate boundary graphs, when the algorithm converges to a good solution. However, there are several weaknesses.

Consistent convergence is difficult to achieve, as the algorithm is highly sensitive to hyperparameter changes and tends to diverge in some settings, however for the solutions that converged, we were able to generate boundary graphs for all class pairs except for the Enzymes dataset. Furthermore, the true expected class probabilities of most solutions do not satisfy the specified stopping criterion, as convergence is determined based on the Monte-Carlo approximation and is therefore subject to the variance of the sampled graphs. These difficulties seem to increase for a growing number of number of classes or larger graphs, but with enough resources, it is still possible to find these solutions. We therefore accept Claim 2, but the effectiveness for a wider range of models remains to be investigated in future work.

When the algorithm does find a good solution, it fails to approximate a significant region of the decision boundary, but instead, the learnt distribution only represents a single region on the boundary. This significantly reduces its value for analysing the decision boundary with the metrics proposed in Claim 3 and explains why in our experiments, the boundary complexity does not seem to be correlated with the different models. This is likely, due to a lack of expressive power in the boundary graph distribution caused by the independence assumption on edges and node features. Based on these findings, we reject Claim 3.

Future research should focus on improving the stability and reliability of training by addressing hyperparameter sensitivity and exploring alternative graph distributions that better capture the full decision boundary. Strengthening convergence consistency would not only improve the algorithm's applicability but also enable a more reliable verification of Claim 4, facilitating a fairer comparison between their novel criterion and standard cross-entropy loss.

### 6.1 What was easy

The architecture of the GNNBoundary is thoroughly explained in the paper, along with most of the essential theoretical concepts needed to understand the algorithm. Each experiment is outlined adequately, with the set-up and the objectives clearly stated in the paper, which makes it easy to follow and assess the claims. Additionally, an official implementation of the algorithm is publicly available through a GitHub repository provided by the authors. The repository also includes model checkpoints as well as demo notebooks that allow for partial reproduction of the experiments, offering a solid foundation for our efforts of replicating this study. The pseudocode for both finding adjacent class pairs and the GNNBoundary was also provided in the paper, which made verifying the given implementation more accessible.

### 6.2 What was difficult

Finding good hyperparameters for GNNBoundary posed particularly challenging, which made it hard to evaluate the effectiveness of their algorithm. Furthermore, we encountered several inconsistencies between the code and the paper. There appeared to be bugs in the novel optimisation criterion, the stopping criterion, and the Monte Carlo approximation used in the GNNBoundary algorithm. Additionally, graph samples were processed differently by the model at test-time, leading to an unpredictable shift in the learnt distribution, and although this was not mentioned in their paper, only the graph edges were sampled from this distribution, while the node features were always assigned the most probable value. Lastly, all datasets were slightly larger than the numbers indicated in their paper, but this did not appear to affect the results in a significant way.

### 6.3 Communication with original authors

We reached out to the original authors for further clarification regarding their paper and the associated code in an effort to resolve the discrepancies and ambiguities previously discussed. However, we did not receive a response.

### Acknowledgments

We thank the FACT-AI team at the University of Amsterdam for their support during this project, and Paul Koole for his collaboration in the early stages. We are particularly grateful to Madhura Pawar for providing guidance and critical feedback.

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

## A  Evaluating boundary margin and misclassification rate

We further evaluate the boundary margin of the adjacent classes and the misclassification rate, to see whether a lower boundary margin results in a lower misclassification rate. We present the results in Table 6.

| Dataset | Class | Adjacent Class | Boundary Margin | Misclassification | Correct Relative Misclassification Prediction from Boundary Margin |
|---|---|---|---|---|---|
| Motif | House | HouseX | 0.48 | 27 | |
| | | Comp4 | 2.80 | 194 | |
| | HouseX | House | 0.57 | 12 | |
| | | Comp5 | 0.93 | 44 | |
| Collab | High Energy | Condensed Matter | 3.21 | 109 | |
| | | Astro | 3.78 | 194 | |
| | Condensed Matter | High Energy | 2.91 | 161 | ✓ |
| | | Astro | 3.61 | 28 | |
| | Astro | High Energy | 4.41 | 223 | ✓ |
| | | Condensed Matter | 5.09 | 21 | |
| Enzymes | EC6 | EC1 | 12.17 | 4 | |
| | | EC2 | 31.03 | 8 | |

Table 6: Comparison of boundary margin and misclassification for adjacent classes. Final column indicates whether the boundary margin between two adjacent accurately predicts the relative misclasification rate.

From the results, there does not seem to be a correlation between boundary margin and misclassification rate. For every dataset except Collab, the misclassification rate is higher for adjacent classes with larger boundary margins, and out of the 6 pairs of adjacent classes, the boundary margin correctly predicts the relative misclassification rate of 2 pairs.

## B  Hyperparameters

### B.1  Training GNN

| Category | Parameter | GCN | | | | GAT |
|---|---|---|---|---|---|---|
| | | Motif | Collab | Enzymes | Reddit | Motif |
| Optimization | Optimizer | SGD | SGD | SGD | SGD | SGD |
| | lr | 0.001 | 0.001 | 0.0001 | 0.004 | 0.001 |
| | Num epoch | 128 | 1024 | 4096 | 100 | 128 |

Table 7: Hyperparameters used to train GNN for each dataset across different architecture. After training for the set number of epochs, the best model is chosen as the model with the highest validation accuracy.

### B.2  GNNBoundary

The equations associated with the hyperparameters are shown for additional clarity.

**Objective function**

$$\mathcal{L}(G) = \sum_{b' \notin \{c_1, c_2\}} \beta \cdot \eta(G)_{b'} \cdot p^*(b'|G)^2 - \sum_{b \in \{c_1, c_2\}} \alpha \cdot \eta(G)_b \cdot (1 - p^*(b|G))^2 \cdot \mathbb{1}_{p^*(b) < \max_{c \in [1,C]} p^*(c)} \quad (5)$$

Here the predicted probability for a Graph $G$ and class $b$ is given as $p*(b|G) = \mathrm{softmax}(\eta(G))_b$.

**Near-Boundary Criterion**

$$\Psi(G) = \mathbb{1}_{p(c_1), p(c_2) \in [p_{\min}, p_{\max}]}(G) \quad (6)$$

**Regularization**

$$R_{\mathrm{budget}} = \mathrm{Softplus}\left(\|\operatorname{sigmoid}(\mathbf{\Omega})\|_1 - B\right)^2 \quad (7)$$

**Dynamic Regularization Scheduler**

$$w_{\mathrm{budget}}^{(t)} = w_{\mathrm{budget}}^{(t-1)} \cdot s_{\mathrm{inc}}^{\mathbb{1}\left\{\Psi\left(G^{(t)}\right)\right\}} \cdot s_{\mathrm{dec}}^{\mathbb{1}\left\{\neg\Psi\left(G^{(t)}\right) \wedge \left(s_{\mathrm{dec}} \cdot w_{\mathrm{budget}}^{(t-1)} \geq w_{\mathrm{budget}}^{(0)}\right)\right\}} \quad (8)$$

| Category | Parameter | GCN | | GAT | Description |
|---|---|---|---|---|---|
| | | Original Datasets | Binary Reddit Dataset | Motif Dataset | |
| Criterion | $\alpha$ | 1 | 1 | 1 | See Eq. 5 |
| | $\beta$ | 1 | 1 | 1 | See Eq. 5 |
| | $W_c$ | 25 | 25 | 25 | Criterion weight (5) |
| Regularisation | L1 | 1 | 1 | 1 | L1 regularisation weight |
| | L2 | 1 | 1 | 1 | L2 regularisation weight |
| | B | 10 | 100 | 30 | Anticipated max edges (7) |
| | $\beta_{\mathrm{softplus}}$ | 1 | 1 | 1 | Controls sharpness of Softplus transition (7) |
| Optimisation | Optimiser | SGD | SGD | SGD | - |
| | lr | 1 | 1 | 0.01 | Learning rate of optimiser |
| | $[p_{\min}, p_{\max}]$ | [0.45, 0.55] | [0.45, 0.55] | [0.45, 0.55] | Relaxed near boundary criterion (6) |
| | $B_{\max}$ | 60 | 400 | 60 | Max edges for convergence |
| | $w_{\mathrm{budget}}^{(0)}$ | 1 | 1 | 1 | Initial budget penalty weight (8) |
| | $s_{\mathrm{inc}}$ | 1.15 | 1.15 | 1.15 | Budget penalty increment (8) |
| | $s_{\mathrm{dec}}$ | 0.98 | 0.98 | 0.98 | Budget penalty decrement (8) |
| | $\mathrm{K}_{\mathrm{MC}}$ | 32 | 32 | 32 | Monte Carlo sample count |
| Graph distribution | $N_{\max}$ | 25 | 150 | 50 | Max number of nodes |
| | $\tau$ | 0.15 | 0.15 | 0.15 | Concrete distribution temperature |

Table 8: Hyperparameters used for GNNBoundary across different datasets. The regularisation parameters, optimisation settings, and graph distribution constraints are listed for GCN and GAT models. The original dataset consists of Motif, Collab, and Enzymes.

## C   Results from our own trained model

We applied the GNNBoundary algorithm to our independently trained GNN model using the same datasets as the original study to assess its effectiveness beyond the provided checkpoints. As shown in Table 9, GNNBoundary successfully generates improved boundary graphs compared to the baseline. However, for the Collab and Enzymes datasets, the generated boundary graphs deviate from the target class probability of 0.5.

Consistent with our reproduced results using the original checkpoint, the standard deviation of class probabilities in the converged runs remains high, and GNNBoundary exhibits a low convergence rate for the Enzymes dataset (see Table 11).

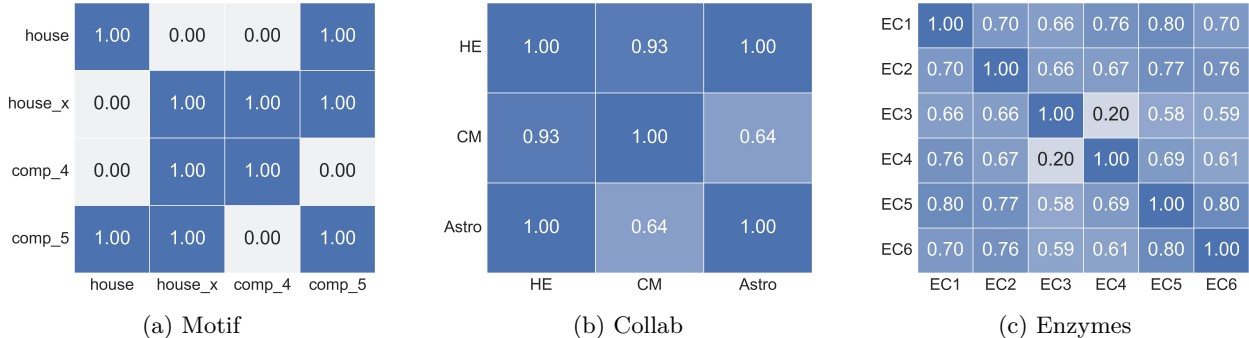

(a) Motif      (b) Collab      (c) Enzymes

Figure 4: Adjacency scores for our retrained models on Motif, Collab, and Enzymes.

| Dataset | Class Pair | | Complexity | GNNBoundary | Baseline |
|---|---|---|---|---|---|
| | $c_1$ | $c_2$ | | $p(c_1)$ / $p(c_2)$ | $p(c_1)$ / $p(c_2)$ |
| Motif | House | Comp5 | 0.194 | 0.521 (0.053) / 0.479 (0.053) | 0.022(0.107) / 0.977 (0.107) |
| | HouseX | Comp4 | 0.341 | 0.502 (0.045) / 0.496 (0.044) | 0.041(0.167) / 0.959 (0.167) |
| | HouseX | Comp5 | 0.509 | 0.505 (0.045) / 0.486 (0.043) | 0.945(0.207) / 0.055 (0.207) |
| Collab | HE | CM | 0.217 | 0.448 (0.027) / 0.498 (0.029) | 0.845 (0.234) / 0.138 (0.231) |
| | HE | Astro | 0.135 | 0.464 (0.017) / 0.459 (0.017) | 0.384 (0.427) / 0.603 (0.439 |
| Enzymes | EC1 | EC5 | 0.262 | 0.480 (0.101) / 0.394 (0.106) | 0.219 (0.362) / 0.333 (0.362) |
| | EC5 | EC6 | 2.87e-14 | 0.422 (0.096) / 0.457 (0.084) | 0.356 (0.417) / 0.221 (0.350) |

Table 9: GNNBoundary results per dataset and class pair from our trained model. For class probabilities, we report the mean and standard deviation over sampled graphs using samples from the best converging run for GNNBoundary. All values are rounded to three decimal places.

| Dataset | Class Pair | | Convergence | Run Means | Run Std. Devs. |
|---|---|---|---|---|---|
| | $c_1$ | $c_2$ | | $p(c_1)$ / $p(c_2)$ | $p(c_1)$ / $p(c_2)$ |
| Motif | House | Comp5 | 0.884 | 0.511 (0.042) / 0.489 (0.042) | 0.162 (0.054) / 0.162 (0.054) |
| | HouseX | Comp4 | 0.410 | 0.501 (0.038) / 0.495 (0.038) | 0.116 (0.066) / 0.119 (0.066) |
| | HouseX | Comp5 | 0.607 | 0.482 (0.042) / 0.487 (0.041) | 0.167 (0.060) / 0.145 (0.064) |
| Collab | HE | CM | 0.973 | 0.459 (0.014) / 0.460 (0.016) | 0.062 (0.009) / 0.077 (0.014) |
| | HE | Astro | 0.004 | 0.455 (0.006) / 0.455 (0.004) | 0.023 (0.004) / 0.020 (0.003) |
| Enzymes | EC1 | EC5 | 0.092 | 0.472 (0.050) / 0.452 (0.053) | 0.190 (0.054) / 0.202 (0.054) |
| | EC5 | EC6 | 0.040 | 0.380 (0.026) / 0.533 (0.036) | 0.206 (0.026) / 0.210 (0.026) |

Table 10: GNNBoundary results per dataset and class pair from our own trained model. The *Convergence* column represents the reproduced value. The columns *Run Means* and *Run Std. Devs.* display the mean and standard deviation of graph samples per run in the format 'mean (std)'. All values are rounded to three decimal places.

## D  Boundary evaluation metrics

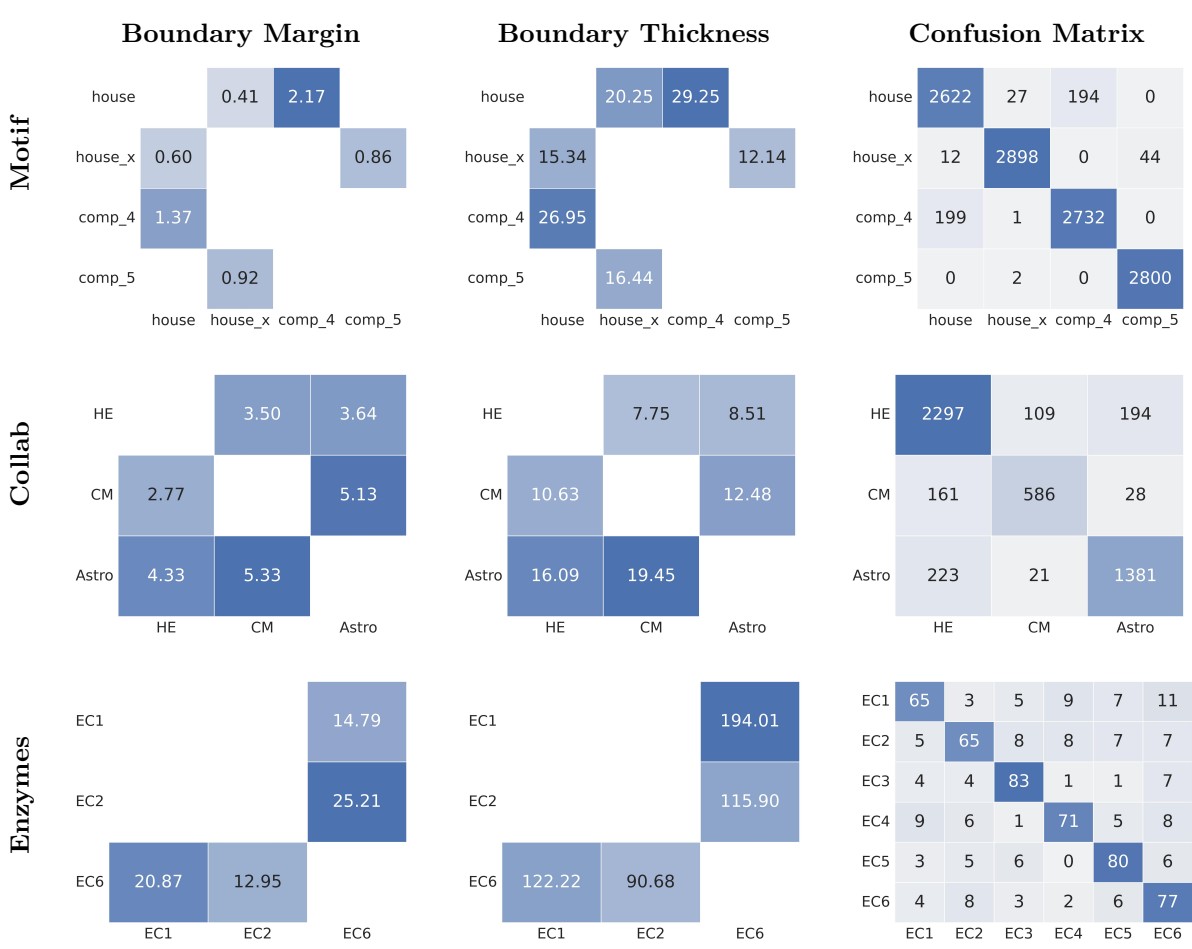

Figure 5: Boundary analysis results for the three original datasets. We compute boundary margin and thickness for adjacent class pairs as well as the confusion matrices for the corresponding models.

# E    Results of GNNBoundary on GAT model

| Dataset | Class Pair | | Complexity | Run Means | Run Std. Devs. |
|---------|------------|----|------------|-----------|----------------|
| | $c_1$ | $c_2$ | | $p(c_1)$ / $p(c_2)$ | $p(c_1)$ / $p(c_2)$ |
| Motif | House | HouseX | 0.050 | 0.506 (0.072) / 0.483 (0.073) | 0.294 (0.150) / 0.295 (0.151) |
| | HouseX | Comp5 | 0.021 | 0.491 (0.077) / 0.507 (0.077) | 0.272 (0.132) / 0.273 (0.132) |
| | HouseX | Comp4 | 0 | $-$ / $-$ | $-$ / $-$ |
| | Comp4 | Comp5 | 0 | $-$ / $-$ | $-$ / $-$ |

Table 11: GNNBoundary results per dataset and class pair from our own trained GAT model. The columns *Run Means* and *Run Std. Devs.* display the mean and standard deviation of graph samples per run in the format 'mean (std)'. All values are rounded to three decimal places.

# F    Graph distribution parameter analysis

Here, we show the parameter distributions of the graph distributions learnt by GNNBoundary. Figure 7 is the parameter distribution for the Motif dataset, Figure 6 for the Collab dataset, and 8 for the Enzymes dataset.

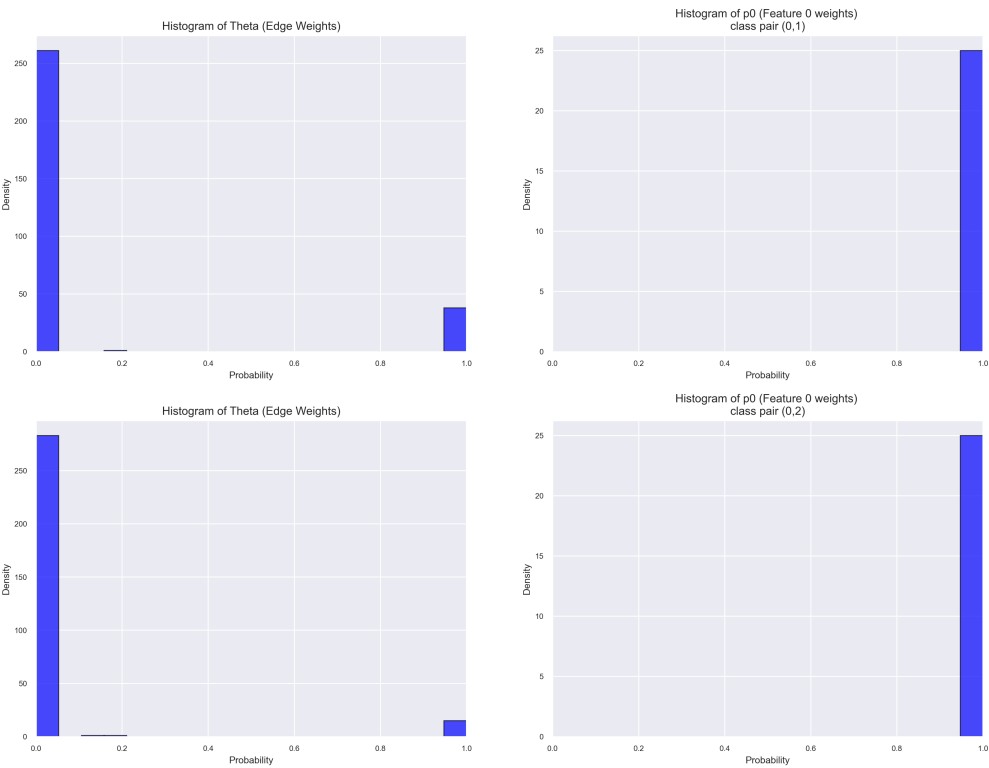

Figure 6: Histogram of graph distribution parameters learnt for the Collab Dataset for all adjacent class pairs.

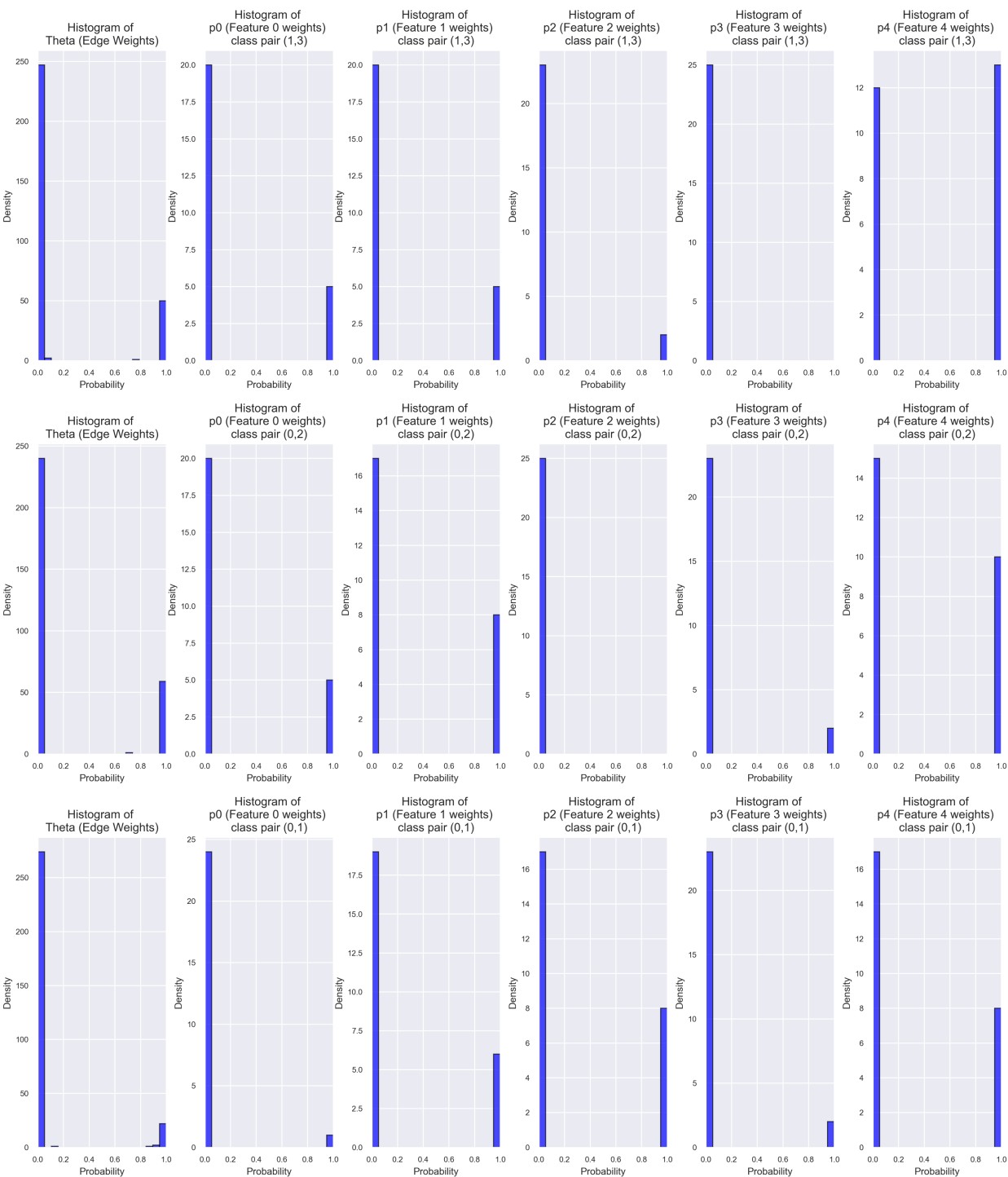

Figure 7: Histogram of graph distribution parameters learnt for the Motif Dataset for all adjacent class pairs.

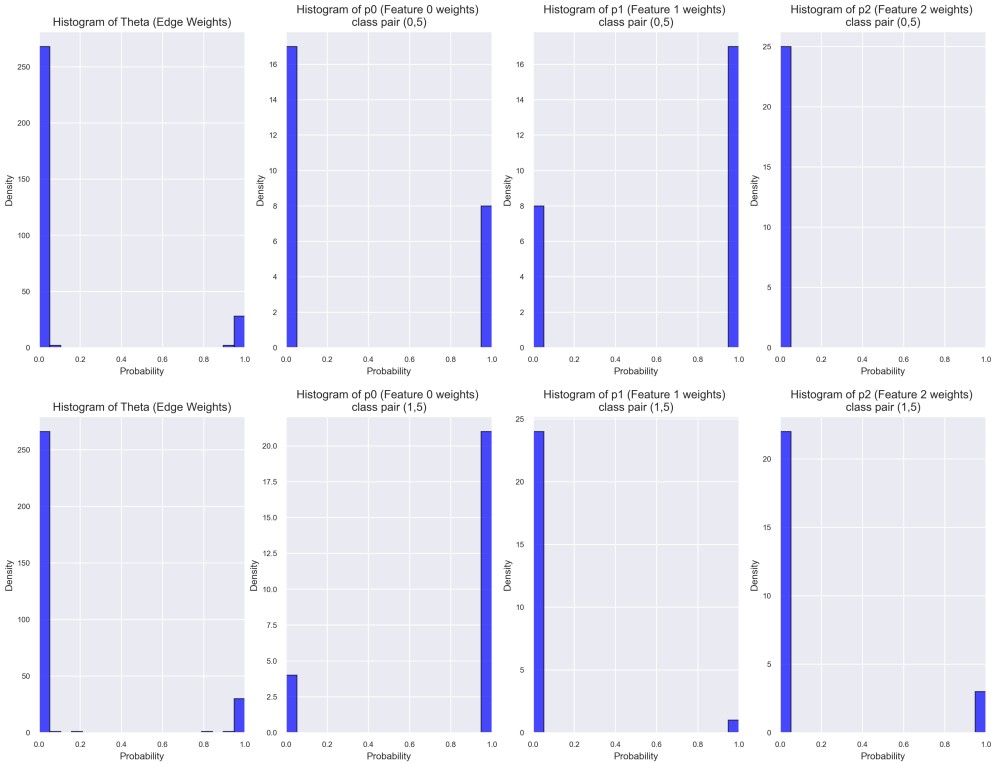

Figure 8: Histogram of graph distribution parameters learnt for the Enzymes Dataset for all adjacent class pairs.

## G GNNBoundary samples

In this section, we visualise the boundary graphs generated by GNNBoundary alongside the graphs obtained from the dataset. We sample 5 graphs using GNNBoundary for each adjacent class pair in the Motif, Collab and Enzymes datasets(see Figure 9).

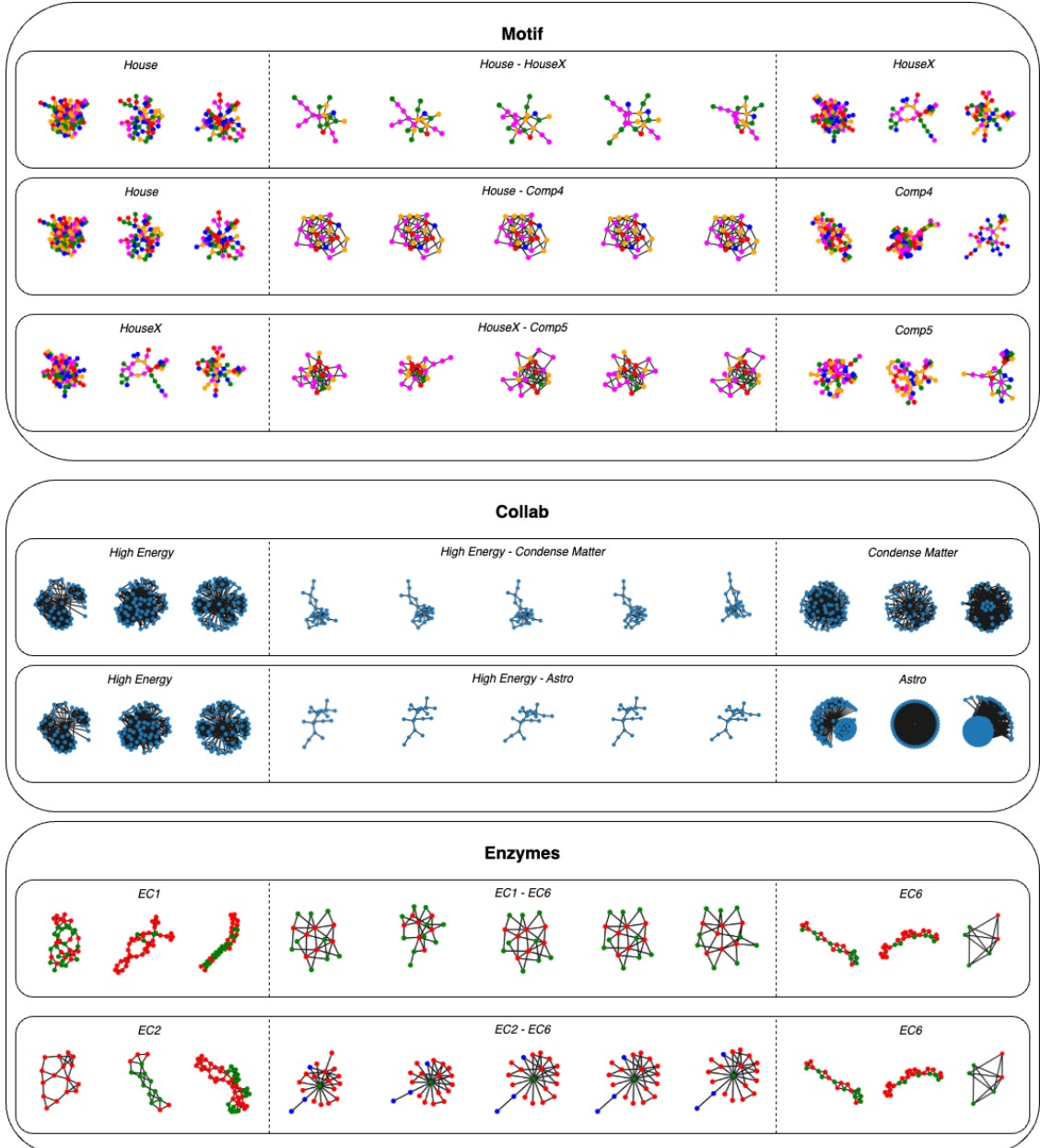

Figure 9: Samples generated by GNNBoundary for each adjacent class pair, alongside the graphs sampled from the dataset, in Motif, Collab, and Enzymes.

