# OpenReview forum: "[RE] GNNBoundary: Towards Explaining Graph Neural Networks through the Lens of Decision Boundaries"
_TMLR — Accepted by TMLR_

### Review · Reviewer_B3Up · 2025-03-08

**Summary Of Contributions:**

This paper analyzes and evaluates the GNNBoundary method, which provides model-level explanations for Graph Neural Networks (GNNs). The authors assess the reproducibility of experiments, the method's robustness, and its applicability to different contexts, identifying both its strengths and limitations.
To achieve this, the authors conduct a series of experiments—some replicating those in the original study and others introducing variations—to both validate certain claims made by the original authors and challenge others. These experiments focus on reproducing the original results, testing the model on a new dataset, evaluating its performance with a different GNN architecture, and conducting a qualitative analysis of the generated graphs concerning the decision boundary.
The findings from these experiments and analyses provide the motivation and scope for this study. Specifically, the reproducibility studies are structured to verify the claims made in the original paper regarding:
- The identification of adjacent class pairs,
- The generation of near-boundary graphs and the method's applicability to any GNN,
- The evaluation metrics used, including boundary margin, thickness, and complexity,
- The proposed loss function’s faster convergence and reduced risk of local minima.

The code is publicly available. Additional results are presented in the Appendix.

**Audience:**

Yes

**Broader Impact Concerns:**

No significant ethical concerns arise from this work. The authors have demonstrated awareness of sustainability aspects by transparently reporting the energy cost of their research. Additionally, no potential negative impacts related to bias, security, or misuse of the proposed technology have been identified.

**Claims And Evidence:**

Yes

**Requested Changes:**

- How challenging was the hyperparameter tuning process? The paper frequently emphasizes the high sensitivity of the procedure to hyperparameter choices. Explicitly quantify this challenge in terms of the number of parameters to be tuned and the associated computational cost. While the list of hyperparameters is provided, specifying the extent of the search required to determine optimal values—such as computational time or resources needed —would help clarify the effort involved.
- In Appendix, Section G, consider adding images of the graphs corresponding to the classes from which the generated graphs originate. Without them, the explanation lacks clarity.
- Style and Grammatical Errors:
    - Sec. 4.2: "We evaluate the GNNBoudnary" → Typo: "GNNBoundary"
    - Sec. 4.1.1: "...might indicate overfitting Guan and Loew (2020)" and Sec. 3.1: "GNNs Scarselli et al. (2009)." → Consider placing citations within parentheses for correctness and readability.

**Strengths And Weaknesses:**

Strengths:
- The analysis and experiments are well-structured and systematically aligned with the claims presented in the original study, ensuring clarity and coherence.
- The work introduces a new dataset and model, extending evaluation beyond what was originally tested
- The experiments and analyses are conducted with a good level of detail, providing both numerical and visual insights that contribute to a deeper understanding of the method’s behavior.
- Discrepancies between the original experiments and the corresponding code are identified and thoroughly examined, with a comprehensive discussion provided in both the main paper and the appendix.
- The submission provides well-founded and compelling evidence to support its statements, offering insights that are valuable to the research community.

Weakness:
- The performance of the Reddit-Binary dataset should be analyzed in greater detail, with a thorough motivation of the results and a clear explanation of the potential underlying factors influencing them, other than the graph-size increment.
- Extending the findings with only a single additional dataset and model may lack robustness, as broader validation across multiple datasets and architectures would provide stronger support for the conclusions.
- GNNBoundary is highly sensitive to hyperparameters, making stable convergence difficult. While the study documents this, a clearer quantification of search effort (e.g., computational cost) would improve clarity.

---

> ### Author Response · Authors · 2025-03-17
> **Response**
>
> Thank you for taking the time to review our submission and provide feedback! We are glad to hear that you find our experiments to provide valuable insights into the method’s behaviour and that our findings are well-supported by our experimental results. We address your concerns below and in our global response.
>
> ### 1. Hyperparameter Search Effort
>
> We could include a brief paragraph quantifying our efforts in tuning the GNNBoundary algorithms. However, we encourage you to refer to our global response, where we outline our reasoning for not conducting a more extensive evaluation of the hyperparameters or the search effort involved.
>
> ### 2. More Datasets & Models
>
> We agree that adding more datasets and models would increase the robustness of our findings. However, we think the addition of the REDDIT-BINARY dataset, and the GAT architecture already explores the applicability of GNNBoundary to more complex domains, and more value would be gained with additional settings being explored in the form of a survey, rigorously comparing competing methods on a set of benchmarks dataset. As we were more interested in exploring the properties of the method rather than doing a simple benchmark, we believe this goes beyond the scope of this paper.
>
> ### 3. Graph Sample Clarity & Style
>
> Thank you for pointing out the style and grammatical errors, as well as our section with generated graphs lacking clarity! We will change this in our revised manuscript.

---

### Review · Reviewer_3mPi · 2025-03-12

**Summary Of Contributions:**

The paper evaluates the GNNBoundary algorithm, a model-level explainability tool designed to interpret Graph Neural Networks (GNNs) through decision boundary analysis. The authors attempt to replicate and extend the experiments of the original paper to assess reproducibility, robustness, and practical applicability.
They focuses and try to verify the following claims on the original paper:
1)GNNBoundary identifies adjacent class pairs within the GNN embedding space. (Partially verified)
2)It generates near-boundary graphs applicable across different GNN models and datasets. (Verified with caveats)
3)These graphs allow for effective decision boundary analysis using boundary margin, thickness, and complexity metrics. (Not verified, with evidence contradicting the claim)
4)An adaptive loss function improves convergence and reduces local minima. (Partially verified)
They also extend the original paper by testing the model on an additional dataset (Reddit dataset) and on an additional architecture (Graph Attention Networks GAT) and by investigating the limitations of the proposed method.

**Audience:**

Yes

**Broader Impact Concerns:**

There are no concerns regarding the ethical implications of this work that would require adding or modifying a Broader Impact Statement.

**Claims And Evidence:**

Yes

**Requested Changes:**

A proper study, such as a grid search or an ablation study, would quantify the exact impact of hyperparameters on reproducibility. The paper attributes performance differences to hyperparameter choices but does not systematically analyze their influence. A structured sensitivity analysis would help determine which parameters have the most significant effect on convergence and accuracy. Additionally, the datasets used in this study appear to be updated versions of those in the original paper. The authors mention that dataset sizes are slightly higher than those reported in Wang and Shen (2024), likely due to updates. This discrepancy can affect the reproducibility of results, as even minor variations in dataset composition could lead to different adjacency relationships and decision boundaries. Since reproducibility studies aim to assess whether results can be replicated under the same conditions, it is important to clarify the potential impact of dataset updates. Running experiments on older versions of the datasets, if available, or quantifying the effect of dataset changes on the results would strengthen the claims about reproducibility.
Both of these aspects could strengthen the assumptions and narrative of the paper. A sensitivity analysis of hyperparameters would provide a more precise evaluation of their role in shaping the results, reinforcing the reliability of the conclusions. Similarly, addressing dataset version differences would enhance the clarity of the reproducibility discussion, ensuring that variations in results are correctly attributed.

**Strengths And Weaknesses:**

Weaknesses:
The paper frequently attributes differences in results to hyperparameter choices but does not provide a systematic sensitivity analysis to confirm this. Additionally, it does not test how random seeds, weight initialization, or stochastic optimization methods influence results. Running multiple trials with fixed vs. varying seeds would help separate systematic errors from random variations. Another limitation is that the datasets used are likely updated versions of those in the original study, which could affect reproducibility. Even small changes in dataset composition may alter adjacency relationships and decision boundaries, making it unclear whether discrepancies arise from methodological differences or dataset updates.

Strengths:
The paper systematically evaluates the original claims by replicating and extending experiments. Testing on an additional dataset, Reddit-Binary, allows for an assessment of scalability, while applying GNNBoundary to a different architecture (GAT) provides insights into its generalizability.
The study also includes robustness checks on convergence and decision boundary metrics, improving the evaluation of GNNBoundary’s reliability. By identifying key limitations the paper provides a balanced analysis and offers directions for improvement.

---

> ### Author Response · Authors · 2025-03-17
> **Response**
>
> Thank you for taking the time to review our submission. We appreciate your feedback and are glad to hear that you find that our additional experiments explore important aspects of the method’s behaviour, and improve the evaluation of its limitations. We address your concerns below and in our global response.
>
> ### 1. Hyperparameter Impact & Sensitivity Analysis
>
> Please refer to the global response, we explain our reasoning for refraining from a more thorough evaluation of the hyperparameters.
>
> ### 2. Dataset Discrepancies
>
> We agree that the differences between the dataset versions might affect reproducibility and this is an unfortunate circumstance. We use the datasets linked in the repository of the official implementation, which seem to match the official source at “https://chrsmrrs.github.io/datasets/”, and to the best of our knowledge, there are no previous versions available. Furthermore, we use the model checkpoints provided by Wang & Shen, which are hopefully trained on the originally reported version. As discussed in the “What was hard” section, we also tried contacting the authors in hopes of addressing this concern, but we did not get any response by the time of writing this comment.
>
> In any case, we think that variations in the reproducibility mostly stem from different model checkpoints, which would likely vary even if we retrained them on the same version of the datasets, e.g. due to the initialisation. As both the class adjacency and the GNNBoundary algorithm (which doesn’t actively use the dataset at all during training) are mainly dependent on the decision landscape learnt by the model, we believe this should outweigh any discrepancies caused by the additional data points. If you still think the effect of dataset changes should be further explored, could you please clarify how we can quantify this most effectively?
>
> ### 3. Random Seeds & Initialisation
>
> Of course, we agree that it is very important to account for stochastic variations. Therefore, we conduct 1000 training runs of GNNBoundary with different random states and sample 500 graphs per run. Following Wang & Shen we use only the best run for evaluation, but in contrast, we also evaluate the variation across runs. Due to the nature of the algorithm, the results can change fundamentally for different model checkpoints (such as obtaining different class adjacencies), which makes it hard to quantify variations in the final class probabilities, and is why we stick to using one model, as in the original paper. However, we still include the results for our retrained checkpoints in the appendix, which should give an idea of the robustness across models.

---

### Review · Reviewer_K7rc · 2025-03-13

**Summary Of Contributions:**

The authors present a reproducibility study of "GNNBoundary: Towards Explaining Graph Neural Networks through the Lens of Decision Boundaries", which introduced a model-level explainability method for GNNs.
They reproduce the original experiments, pointing out some reproducibility concerns, and introduce further analyses and tests to verify the original claims.

**Audience:**

Yes

**Claims And Evidence:**

Yes

**Requested Changes:**

1. More important concerns:
    - The baseline used in the original paper is admittedly quite naive, isn't there a way to improve it?
    - The authors decided to sample graphs directly from the dataset instead of using GNNInterpreter (paragraph 4.5). They say to do so to verify that the method is independent from the sampling source, but I could not find this claim in the original paper, nor it is reported in paragraph 2. To what extent does this choice explain the difference in results?
    - Is there a reason why it was not possible to report a standard deviation for the boundary complexity in Table 3?
    - In the paragraph "Boundary analysis", it is claimed that there is no correlation between the boundary margin and the misclassification rate. Can the authors quantify this correlation?
    - In the same paragraph, the authors say that the claim that boundary thickness is a measure of robustness to adversarial perturbations would need to be explored in a future work. In my opinion, having two reproducibility studies for the same method is a bit hard to justify, and it would have been interesting to see this point explored further.
    - In paragraph 5.2, the authors show that what GNNBoundary finds is a small portion of the real boundary. Does this result change between runs?


2. Some concerns that I have on the original paper that were not analysed in this review:
    - Is there a meaningful difference between the boundary margin and the boundary thickness? They both measure a distance between a class and a boundary, I am left wondering if they are redundant.
    - The original paper takes the definition of boundary complexity from a paper by Guan and Loew (2020), who argue that this metric doesn't take into account if a boundary shows more complex local structures, e.g. zigzags. It would have been interesting to criticise and analyse the effects of this simplification.

3. Minor mistakes or typos:
    - The notation used for GNN is a bit confusing, where $l$ seems to indicate both the number of GNN embedding functions, but also a specific layer
    - $D^{(l)}$ has not been defined
    - In paragraph 3.1, $G\in\mathcal{R}^{(L)}_{c}$ should be $G\in\mathcal{R}^{(L)}_{\hat c}$
    - In equation (2), the $dt$ is missing from the integral. Also, the distribution $P$ has not been introduced.
    - When introducing the metric "boundary thickness", the authors describe it as "the expected distance traveled along line segments
 between classes across a decision boundary". I don't agree with this description because the formula computes an expected value between class $c_1$ and the boundary $c_1 || c_2$, not between $c_1$ and $c_2$.

**Strengths And Weaknesses:**

Strengths:
- The paper is well structured. I particularly appreciate paragraph 2, where the authors provide a schematic summary of the claims tested, which is not present in the original paper.
- The authors don't limit their work to just reproducing the original experiments; they also allegedly (I didn't verify this myself) extended the original code, they provide further experiments on one more dataset and one more model, and they give further insights on the original results.

Weaknesses:
- The description of the original method is sometimes a bit short, forcing the reader to look at the original paper too.
- In some parts of the paper, it is not fully clear what is a new contribution and what is not. For example, in paragraph 4.1 the authors frequently use the pronoun "we", where it seems they are describing the original method.
- It would have been nice to add more datasets and models.
- See in the section below some other limitations.

---

> ### Author Response · Authors · 2025-03-17
> **Response**
>
> Thank you for your thoughtful review and detailed feedback. We truly appreciate the time and effort you put into it. We will address each point from your comment in order:
>
> ### Important concerns
>
> - Yes, we admit that the baseline proposed in the original method does perform poorly and we also find this to be an issue during our evaluation. We tried to look for alternatives but could not come up with any promising ideas, and as mentioned by Wang & Shen, due to the novelty of their algorithm they were also unable to find an additional baseline from existing literature. However, since GNNBoundary consistently generates graphs that meet the target probability criterion of 0.45 to 0.55 in most scenarios, we believe that introducing another baseline for comparison would not add significant value.
> - In the second paragraph of section 5.2 of the original GNNBoundary paper, you can find Wang and Shen’s claim that GNNBoundary should achieve the same result regardless of the sampling method used. Alongside the already provided motivation for this design choice we also had two other concerns that explain why we chose to sample directly from the dataset:
>   1. As mentioned in our paper, both the dataset and the model checkpoints provided in the official implementation of GNNBoundary seem to be different from what was reported in the original paper. Therefore, training the GNNInterpreter from scratch on these new versions of datasets and models could introduce even more variance and might hinder the reproducibility of the experiments.
>   2. The authors motivated using GNNInterpreter as a sampling source because they wanted to explore generating the boundary graphs without having full access to the initial dataset on which the model was trained. However, this reasoning seems unclear, as GNNInterpreter directly utilizes embeddings from the dataset during training, so although GNNBoundary will only indirectly sample from the original dataset via GNNInterpreter, one will still need full access to the dataset to obtain an instance of GNNInterpreter.
>   3. GNNinterpreter generates the most discriminative graphs for the decision region learnt by a model. Therefore we would expect the distribution of the data generated from this to be different from the true data distribution of the decision region, where we would expect more noise and variation.
> - We did not provide the variance for the complexity because we were trying to reproduce the tables from the original paper which also did not include it. We will add the variance if you think it will help with the completeness of our results, however, since we showed that these metrics cannot be accurately computed using GNNBoundary, we believe adding the variance will not give additional insight into the results.
> - We will better quantify the correlation (or rather lack thereof) between the boundary margin and the misclassification rate in our revised manuscript. We could either calculate the correlation or provide specific examples where this rule is not upheld. Which approach do you believe is more appropriate to clear up this point?
> - The original authors claim that the boundary thickness is a measure of robustness to adversarial perturbations based on existing literature where this phenomenon was explored for other types of networks (not GNNs). We agree that this would be valuable to explore in our setting, but since we show that the thickness metric can not accurately be computed using only the GNNBoundary sampler, we lack the basis for further experiments.
> - We will add additional visualizations comparing the results across different runs in a revised version of our paper.
>
>
> ### Concerns on the original paper
>
> - Based on our review of the literature, margin, and thickness are distinct metrics as they measure different aspects. Margin quantifies the smallest distance between the decision boundary and a data point, while thickness refers to the width of the decision boundary region, which reflects how gradually the class probabilities change as data points transition from one class to another. Since both metrics provide insights into why certain types of misclassification errors may be more frequent than others, we acknowledge that they may sometimes appear redundant, but we think that an in-depth analysis of a model could benefit from both metrics.
> - Further exploring the complexity metric is an interesting idea and we thank you for this suggestion. We believe this would have be worth exploring, however, as for boundary thickness, the GNNBoudnary samples are not able to represent the whole decision boundary so the suggested metrics can not be computed accurately which hinders our ability to conduct further meaningful experiments.
>
> ### Minor concerns
>
> - We will make all the suggested changes and provide a revised version of our paper as soon as possible.

---

### Author Response · Authors · 2025-03-17
**Global Response**

We thank all reviewers for taking the time to review our paper and for their thoughtful suggestions.

### Hyperparameter Evaluation

We want to address the reviewers’ concerns regarding a hyperparameter search and an evaluation of their influence. Since we do not have access to all of the hyperparameter values used in the original paper, and running GNNBoundary for different trained model checkpoints might generally require different hyperparameters, it would be best to conduct a thorough hyperparameter search, maximising the reproduced performance for ideal comparability. We struggled with finding a productive way of doing this in the GNNBoundary setting. The first challenge is the amount of parameters directly influencing the training in this unique optimization setting, which extends beyond the parameters of the theoretical algorithm in the implementation. Additionally, the uniqueness of the problem leaves us with little intuition or theoretical motivation for choosing viable ranges to explore. Secondly, it is hard to define a concrete goal for the hyperparameter search. During manual experimentation, we found that decreasing the learning rate can stabilise the learning process by reducing the impact of fluctuations due to the Monte Carlo approximation. This should intuitively improve the convergence rate, but at the same time, it led to worse solutions being found overall. Furthermore, as we mainly utilize the best of 1000 runs for evaluation and observe considerable variance across runs, using the best hyperparameter settings based on an estimate with a significantly lower number of runs is not guaranteed to improve the final performance. We think, running a structured sensitivity analysis would run into the same issue, and conducting a search with the full 1000 runs per combination is not feasible for us. Otherwise providing an estimate of the computational requirements for this search seems unfair to us, as in a practical setting this would not be required.

We are open to discussing further suggestions on how to circumvent this, and will otherwise include a more extensive reasoning in the revised manuscript.

---

### Comment · Action_Editor_VphM · 2025-03-27

Dear reviewers,

Just a reminder that the discussion phase will close very soon. Please ensure you have all the information you need to submit the final recommendation, and feel free to ask further questions to the authors should this not be so.

Thanks,
The AC

---

### Decision · Action_Editor_VphM · 2025-04-10

**Recommendation:** Accept as is

**Comment:**

The paper had three reviews, which were positive in terms of validity and completeness of the claims. There is a consensus that the paper is written well, and most sections are interesting, including the novel experiments and studies. Two reviewers (B3Up, 3mPi) were concerned by the lack of a systematic investigation of hyper-parameters. However, as the authors clarify, this is much harder when considering explainability, and both reviewers were satisfied by the answer. All further comments by reviewer K7rc were addressed. The methodological part is difficult to read without having read the original paper, but this is expected for a reproducibility study.

The authors are invited to double check and proofread everything, as there are minor typos (e.g., "Recommendation systems Wu et al. (2022b)" has the citation formatted wrongly, "explainability methods(Luo et al. (2020) [...])" has a missing space, etc.).

**Audience:**

The paper is targeted to researchers interested in explainability of graph neural network models, which is an active topic. The paper focuses on a very important aspect of this topic, namely, the difficulty of selecting valid hyper-parameters and ensuring reproducibility due to the complexity of defining valid benchmarks and metrics in the field. As such, the paper is valuable for the community even beyond the specific re-implementation.

**Claims And Evidence:**

The paper is a reproducibility study of the GNNBoundary paper published at ICLR 2024. The authors thoroughly investigated all claims from the original paper, with only one being fully reproducible due to potential variations in hyperparameters. They also found missing elements in the original code, and discrepancies in the provided checkpoints. Furthermore, they extended the analysis to one additional dataset (Reddit) and one additional model (GAT). Experiments are comprehensive and they provide clear and convincing evidence for all claims.